# The Functional Characterization of *DzCYP72A12-4* Related to Diosgenin Biosynthesis and Drought Adaptability in *Dioscorea zingiberensis*

**DOI:** 10.3390/ijms24098430

**Published:** 2023-05-08

**Authors:** Weipeng Wang, Lixiu Hou, Song Li, Jiaru Li

**Affiliations:** State Key Laboratory of Hybrid Rice, Department of Plant Science, College of Life Sciences, Wuhan University, Wuhan 430072, China

**Keywords:** *Dioscorea zingiberensis*, diosgenin, Sterol C26-hydroxylase, CYP72A, drought stress

## Abstract

*Dioscorea zingiberensis* is a perennial herb famous for the production of diosgenin, which is a valuable initial material for the industrial synthesis of steroid drugs. Sterol C26-hydroxylases, such as *Tf*CYP72A616 and *Pp*CYP72A613, play an important role in the diosgenin biosynthesis pathway. In the present study, a novel gene, *DzCYP72A12-4*, was identified as C26-hydroxylase and was found to be involved in diosgenin biosynthesis, for the first time in *D. zingiberensis*, using comprehensive methods. Then, the diosgenin heterogenous biosynthesis pathway starting from cholesterol was created in stable transgenic tobacco (*Nicotiana tabacum* L.) harboring *DzCYP90B71*(QPZ88854), *DzCYP90G6*(QPZ88855) and *DzCYP72A12-4*. Meanwhile, diosgenin was detected in the transgenic tobacco using an ultra-performance liquid chromatography system (Vanquish UPLC 689, Thermo Fisher Scientific, Bremen, Germany) tandem MS (Q Exactive Hybrid Quadrupole-Orbitrap Mass Spectrometer, Thermo Fisher Scientific, Bremen, Germany). Further RT-qPCR analysis showed that *DzCYP72A12-4* was highly expressed in both rhizomes and leaves and was upregulated under 15% polyethylene glycol (PEG) treatment, indicating that *DzCYP72A12-4* may be related to drought resistance. In addition, the germination rate of the diosgenin-producing tobacco seeds was higher than that of the negative controls under 15% PEG pressure. In addition, the concentration of malonaldehyde (MDA) was lower in the diosgenin-producing tobacco seedlings than those of the control, indicating higher drought adaptability. The results of this study provide valuable information for further research on diosgenin biosynthesis in *D. zingiberensis* and its functions related to drought adaptability.

## 1. Introduction

Diosgenin, a well-known steroidal sapogenin, is not only an important and valuable material for the production of many steroid drugs [1], but it is also of great pharmaceutical value itself [2,3,4,5,6]. Dioscin is the main form of diosgenin in plants, composed of the diosgenin (27 Carbon atom-formed steroidal skeleton) with an oligosaccharide at the C (3) hydroxy group. The water solubility and efficacy of dioscin are often greatly reduced after the hydrolyzation of the side sugar chain. In nature, diosgenin is distributed in many plants, but it is mainly *Dioscorea* plants that can meet the requirements for industrial scale extraction. There are about 137 *Dioscorea* species containing diosgenin, and in more than 30 of them, the content of diosgenin is higher than 1% by dry weight [3]. The highest record is up to 16.15%, found in a single *Dioscorea zingiberensis* rhizoma in China [7].

Previous studies have shown that cholesterol is the main precursor of diosgenin biosynthesis, which has the most consistent 27 Carbon skeleton structure with diosgenin (Figure 1b,c) [6,7,8,9,10,11,12]. It is generally recognized that the basic 17 carbon atom-formed perhydrocyclopentanophenanthrene skeleton (Figure 1a) is synthesized starting from acetyl coenzyme A(Acetyl-CoA). Acetyl-CoA can be converted to 2,3-oxide squalene through the mevalonate acid (MVA) pathway or the methylerythritol phosphate (MEP) pathway. Subsequently, 2,3-oxide squalene is led into the cholesterol biosynthesis pathway by cycloartenol synthase (CAS) and finally converted to cholesterol via a variety of enzymes [13,14,15]. Many in-depth studies have been conducted on the metabolism of cholesterol [13] and its derivatives including phytosterols, steroidal glycoalkaloids (SGAs) [16,17] and brassinosteroids [12]. A series of hydroxylation, oxidation and ring formation reactions sequentially occur on the side chain of cholesterol to produce diosgenin, catalyzed by cytochrome P450 monooxygenases (CYP450s) [10,11]. Meanwhile, UDP-dependent glycosyltransferases (UGTs) are involved in the hydroxy group glycosylation, mainly at the C3 position [18,19]. De novo synthesis of diosgenin in yeast from the milligram to the gram scale was achieved through metabolic engineering strategies aimed at the synthesis of cholesterol precursors [20]. The network of cholesterol biosynthesis was also illustrated in *Paris polyphylla*, and a high-efficiency cholesterol synthesis system was established promoting the efficiency of diosgenin production in plant chassis [21].

It has been reported that furostanol I [22], protodioscin and sitosterol [23] are intermediates in diosgenin biosynthesis. Furostanol I has a typical 27-C skeleton with an E ring formed through the bonding of hydroxyls at the C16 and C22 positions (C16-O-C22, Figure 1c). A similar compound has also been found related to SGAs’ biosynthesis starting from cholesterol in *solanaceous* plants. In that process, several CYP450s including GAME7/PGA2, GAME8/PGA1, GAME11 and GAME6 sequentially catalyze the hydroxylation and oxidation of cholesterol at the C22, C26 and C16 positions, and finally form the furostanol-type aglycone [16]. DWF4(CYP90B1) in *Arabidopsis thaliana* [23], CYP90B2 in tomato (*Solanum lycopersicum*) and *Dz*CYP90B71(*D. zingiberensis*) were previously reported as cholesterol C22-hydroxylase [11,24]. Another report demonstrated that *Pp*CYP90G4 in Himalayan paris (*Paris polyphylla*) and *Tf*CYP90B50 in fenugreek (*Trigonella foenum-graecum* L.) encoded sterol C16,22-dihydroxylase in the diosgenin biosynthesis pathway coupled with several C26-hydroxylases including *Pp*CCYP94D108, *Pp*CCYP94D109, *Pp*CCYP72A616, *Tf*CYP72A613 and *Tf*CYP82J17 [10]. *Dz*CYP90G6 in *D. zingiberensis* (monocots) encodes C16-hydroxylase, which is different from its homolog, *Pp*CYP90G4. PGA1 and PGA2 in potato (*Solanum tuberosum*) belong to the CYP72A subfamily, reported to be C26- and C22-hydroxylase, respectively. Clearly, the CYP450s of the CYP72A subfamily play a crucial role in diosgenin biosynthesis. Recently, a chromosome-scale genome (629 Mb) of *D. zingiberensis* was generated [25]. Based on that work, a genome-wide analysis of CYP72A in *D. zingiberensis* was performed. A total of 25 *CYP72A* genes were identified in silico, and nine of them were reported to have high correlation with diosgenin metabolism [26].

Tobacco (*Nicotiana tabacum* L.), as a model plant, containing an endogenous cholesterol biosynthesis pathway, can supply sufficient precursors for diosgenin synthesis [21]. In the meantime, as a eukaryotic organism, tobacco possesses the necessary conditions for cytochrome 450 to exert its activity. Therefore, it is very suitable for the study of CYP450 candidates related to the diosgenin biosynthesis pathway. In fact, the tobacco transient expression system has been utilized to perform high-throughput function characterization of unknown CYP450s and has successfully illustrated the diosgenin biosynthesis pathway in Himalayan paris and fenugreek for the first time [9]. In the present study, a high-throughput function characterization platform was built by expressing a combination of *Dz*CYP90B71(22R-oxidation) and *Dz*CYP90G6 (C16-oxidation) in tobacco, which could biosynthesize C16,22-dihydrocholesterol form cholesterol to supply the substrate for C26-hydoxylase. Using this platform, potential candidates of CYP450s were studied. Finally, the *DzCYP72A12-4* gene was cloned and identified as sterol C26-hydroxylase in *D. zingiberensis*. The amount of diosgenin in the transgenic tobacco plants was determined using an ultra-performance liquid chromatography system (Vanquish UPLC 689, Thermo Fisher Scientific, Bremen, Germany) tandem MS (Q Exactive Hybrid Quadrupole-Orbitrap Mass Spectrometer, Thermo Fisher Scientific, Bremen, Germany) (UPLC–MS/MS). The column equipped on the UPLC system was the Hypersil GOLD C18 column 690 (2.1 mm × 100 mm, 3 m, Thermo Fisher Scientific, Bremen, Germany). Meanwhile, some physiological indexes of the transgenic diosgenin-producing tobacco plants were measured to explore the drought adaptability.

## 2. Results

### 2.1. Sequence Analysis of the CYP72As in D. zingiberensis

According to a previous study, it is inferred that CYP72As function as C26-hydroxylase catalyzing the furostanol end-of-tail hydroxylation in the diosgenin biosynthesis pathway [10]. However, no CYP72As in *D. zingiberensis* have been identified as participating in furostanol hydroxylation so far. A phylogenetic tree was constructed using the neighbor-joining (NJ) method (1000 bootstrap values) (Figure 2) to confine the number of CYP450 candidates involved in diosgenin biosynthesis. In total, 23 previously reported CYP450s (Appendix A) participating in steroid metabolism were exploited as potentials to perform the genome-wide screen utilizing the basic local alignment search tool (BLAST). A total of 47 protein sequences, including 25 members of the CYP72A family described in a previous work [26], were chosen for further analysis (Appendix A).

Based on the phylogenetic analysis, the chosen proteins close to C26-hydroxylase could be divided into three groups (Figure 2), consisting of CYP72As (Clade III), CYP94s (Clade VI&VII) and CYP734s (Clade III). *Dz*CYP94D143 (Clade VI) and *Dz*CYP94N8 (Clade VII) are two characterized C26-hydroxylases in the diosgenin biosynthesis pathway cloned from *D. zingiberensis*, which means screened sequences in this clade might have similar activity with C26-hydroxylase. Although *Tf*CYP82J17 was in clade V, its identities with the other members were relatively low (<50%), which means no specific homolog of *Tf*CYP82J17 existed in *D. zingiberensis*.

As we can see from Figure 2, CYP72As occupied nearly half of the phylogenetic tree, including 24 *Dz*CYP72As (except for *Dz*CYP72A19) as well as three known CYP450s, *Pp*CYP72A616(monocots), GAME7 and *Tf*CYP72A613 (dicots). *Dz*CYP72A sequences were assigned into two clades (Clade I and Clade II, Figure 1) consistent with that described previously [26]. Clade III contained five sequences from *D. zingiberensis* and four reported CYP734As including the well-studied *At*CYP734A1(BAS1, formerly CYP72B1), which is involved in the inactivation of BRs castasterone (CS) and brassinolide (BL) through the carbon 26 hydroxylation [27]. Clade IV comprised three characterized CYP450s (*Dz*CYP90B71, *Dz*CYP90G6 and *Pp*CYP90G4). In addition, GAMEs reported in the SGAs biosynthesis were scattered into different clades and served as functional references. Obviously, most of the screened sequences were *Dz*CYP72As, which may be related to the later C26-oxidation step (also called the end-of-tail hydroxylation) in diosgenin biosynthesis.

Multiple sequence alignment demonstrated that extreme similarity occurred in the nucleotide sequences of *Dz*CYP72A belonging to Clade I (DzCYP1-13/23/24) and Clade II (DzCYP72A18-20), respectively (Appendix A). Pairwise alignment was performed between pairs of *Dz*CYP72As using protein sequences and nucleotide sequences, respectively (Appendix A). *Dz* CYP72A14, *Dz*CYP72A15 and *Dz*CYP72A16 had relatively low similarity with other *Dz*CYP72As, both in the protein and nucleotide sequences. *Dz*CYP72A14 and *Dz*CYP72A15 were relatively close to Clade I, while DzCYP72A16 was close to Clade II. *Dz*CYP72A14 and *Dz*CYP72A25 had the highest similarity to *Tf*CYP72A613 and *Pp*CYP72A616, two characterized C26-hydroxylases. Interestingly, KAJ0962633.1 showed high identity with *Dz*CYP72A17/20/21/22 in the protein sequence but low identity in the nucleotide sequence. In contrast, KAJ0961252.1 showed high identity with *Dz*CYP72A13/23/2/9 in the nucleotide sequence but low identity in the protein sequence.

### 2.2. Gene Clone and Sequence Analysis of DzCYP72A from D. zingiberensis

*DzCYP72A12-4* was amplified using *D. zingiberensis* cDNA as the template based on the analysis above. Nucleic acid sequencing revealed the actual sequence was closest to the predicted DzCYP72A12 (identity: 87.7%). A 53 amino acid residue deletion occurred in the DzCYP72A12-4 protein sequence (Appendix A). Analysis of *DzCYP72A12-4* showed that it encoded a protein of 465 amino acids, with a predicted molecular weight of 53.17 kDa, and a theoretical isoelectric point (pI) of 9.15. The conserved domain sequence (CDS) analysis of DzCYP72A12-4 revealed that it belonged to the cytochrome P450 (CYP) superfamily and contained a putative transmembrane helix region from position 7 to position 29 and a likely PX domain from position 133 to 244.

*Dz*CYP90B71 and *Dz*CYP90G6 were cloned from *D. zingiberensis* and were co-expressed in tobacco to produce the substrate for C26-hydroxylase. The *Dz*CYP90B71 protein catalyzes cholesterol hydroxylation at the C22 position, producing C22-hydroxycholesterol, which is subsequently hydroxylated by the *Dz*CYP90G6 protein producing 16,22-dihydroxycholesterol. *Dz*CYP94N8 was also cloned as a positive control of C26-hydroxylase and co-transformed into tobacco with *Dz*CYP90B71 and *Dz*CYP90G6 to complete the diosgenin heterogenous biosynthesis pathway.

### 2.3. Reconstitution of the Diosgenin Heterogenous Biosynthesis Pathway and the Function Characterization of DzCYP72A12-4 in Tobacco

The diosgenin heterogenous biosynthesis pathway was reconstituted in stable transgenic tobacco harboring a combination of *DzCYP90B71/DzCYP90G6/DzCYP72A12-4* (OE3) or *DzCYP90B71/DzCYP90G6/DzCYP94N8* (positive control, PC). Transgenic tobacco plants harboring *DzCYP72A12-4* (negative control 1, NC1) or *DzCYP90B71/DzCYP90G6* (negative control 2, NC2) were created as background controls. Wild type tobacco was used as the blank control. Genomic DNA PCR was performed for the preliminary screening of positive transgenic plants surviving on MS medium containing 50 ug/mL hygromycin. The expression level of exogenous genes in the selected lines was analyzed by qRT-PCR using the primers listed in Appendix A. Transgenic lines with a relatively high expression of the genes of focus were selected for subsequent experiments.

Ultra-performance liquid chromatography tandem MS was utilized to analyze the metabolic profiling of the tested tobacco leaf extracts, revealing that co-expression of DzCYP90B71, DzCYP90G6 and DzCYP72A12-4 resulted in the heterologous production of diosgenin in tobacco but not in the NC1, NC2 and WT controls (Figure 3). The UPLC–MS/MS results showed that a new peak appeared at 14.54 min (Figure 3A), indicating the formation of a new product with almost the same retention time (14.51 min) and mass spectra (415.32, +H) as those of the diosgenin standard (Figure 3B). This result suggests that DzCYP72A12-4 works as C26-hydroxylase in the diosgenin biosynthetic pathway.

### 2.4. Drought Resistance Increased in the Transgenic Tobacco

The seeds of three OE3 lines and the NC1, NC2 and WT plants were used to test the physiological changes under drought stress. The germination rate and malondialdehyde (MDA) concentration were measured. The germination rate (GR) is a common index of seed vigor. The GRs of all lines were affected by the 15% PEG treatment, but the degree of influence varied with each line (Figure 4A). The seed germination rate of the control group (NC1, NC2 and WT) decreased significantly after the 15% PEG treatment. In contrast to the control group, the GRs of the OE3 lines were also affected but to a lesser extent. The germination rates of the OE3 lines and the control group were significantly different. In addition, the MDA concentration in the OE3 lines did not change as significantly as that in the control group after drought stress (Figure 4B). Under the stress of 15% PEG, the diosgenin-producing tobacco seeds showed a significantly higher germination rate and lower MDA concentration than the controls, suggesting that they had the ability to cope with more severe drought pressure than the controls.

In order to further explore the correlation between the *DzCYP72A12-4* gene and drought stress, analysis of the *DzCYP72A12-4* expression patterns and levels under drought stress in *D. zingiberensis* was conducted. The results showed that the transcript level of *DzCYP72A12-4* was upregulated at first after the treatments of PEG6000 (15%) reaching a peak level around 4 h; then, it subsequently decreased to a level even lower than the normal control after the 96 h PEG treatment (Figure 5). In general, the expression level of the *DzCYP72A12-4* gene in *D. Zingiberensis* increased significantly within 48 h under drought stress. However, after 72 h of PEG treatment, the expression level was significantly lower than the control.

## 3. Discussion

Since the beginning of the steroidal pharmaceutical industry in the first half of the 20th century, diosgenin has become an important starting material for the industrial scale synthesis of steroidal drugs. Of the various sapogenins, diosgenin could be prized because of its suitable structure and abundant source capable of meeting industrial scale demand [28]. Recently, many studies have found that diosgenin itself has therapeutic effects on a variety of diseases, such as cancers [29], inflammatory and metabolic diseases [30], diabetes [5], hypolipidemia [31], atherosclerosis [32] and cardiac diseases [33]. With the increasing use of diosgenin in the pharmaceutical industry and therapeutic treatment, the demand for diosgenin is increasing rapidly. However, the current sources of diosgenin remain the same, and the germplasm resources have not been improved significantly. The rapidly increasing demand for diosgenin is causing great pressure on industrial extraction. At the same time, problems such as environmental pollution and germplasm degradation arise.

Increasing the content of diosgenin biosynthesized in plants can both alleviate the economic pressure faced by the industrial extraction of diosgenin and avoid the policy risks caused by the environmental issues. Great progress has been made in the exploration of the diosgenin biosynthesis pathway, and many cytochrome 450 s, CYP90B, CYP90G, CYP94 and CYP72A, for instance, have been uncovered catalyzing the hydroxylation and oxidation of cholesterol to produce diosgenin [9,10]. As the last step to produce diosgenin, C26-oxidation (also called end-of-tail hydroxylation) could be catalyzed by any one of several distinct CYP450s, such as *Pp*CYP72A616, *Pp*CYP94D108 and *Pp*CYP94D109 in Himalayan paris or *Tf*CYP72A613 and *Tf*CYP82J17 in fenugreek, which reflects the relatively loose pairing rule between substrates and enzymes. However, none of the CYP72As, a large CYP450 subfamily playing crucial roles in catalyzing various sterol biosynthesis reactions, have been identified as diosgenin C26-hydroxylase in *D. zingiberensis*.

Our previous work published a high-quality chromosome-scale genome (629 Mb) of *D. zingiberensis* [25], which revealed that the *DzCYP72A* genes were physically adjacent to each other on chromosomes LG01 and LG09, forming two gene clusters, suggesting the close relationship of *DzCYP72A* in function. The genome-wide analysis in *D. zingiberensis* identified twenty-five *DzCYP72A* genes, and nine of them were found to have a correlation with the content of diosgenin [26]. In addition, *Pp*CYP72A616 and *Tf*CYP72A613 have been characterized as C26-hydroxylase/oxidase for diosgenin biosynthesis; their CYP72A homologs in *D. zingiberensis* could be inferred to have similar catalyzing activity.

Screening specific CYP72As participating in diosgenin biosynthesis is meaningful as well as challenging work, considering the high similarity of these CYP450s in structure and function. The C26-hydroxylase/oxidase related phylogenetic tree was constructed to confine the number of CYP450 candidates involved in diosgenin biosynthesis. Then, 47 protein sequences described in previous work were chosen for further analysis (Appendix A). The phylogenetic analysis showed that the *Dz*CYP72As were divided into two clades (Clade I and Clade II, Figure 2), which corresponded to the two gene clusters mentioned above. Multiple sequence alignment demonstrated that extreme similarity occurred in the coding sequence of the *Dz*CYP72As belonging to Clade I (*Dz*CYP1-13/23/24) and Clade II (*Dz*CYP72A18-20), respectively (Appendix A), suggesting the close relationship of evolution. Pairwise alignment was performed to calculate the identities between each pair of the *Dz*CYP72As using proteins and nucleotides, respectively (Appendix A), revealing that the identities of the nucleic sequences were very high in each gene cluster. In addition, *Dz*CYP72A6/16/17 were reported to have a high correlation with diosgenin metabolism in *D. zingiberensis*.

In order to perform high-throughput function characterization of new CYP450s, the tobacco transient expression system was utilized to explore the diosgenin biosynthesis pathway for the first time in Himalayan paris and fenugreek [9]. Then, more novel diosgenin-biosynthesis-related CYP450s, *Dz*CYP90B71 (22R-oxidation), *Dz*CYP90G6 (C16-oxidation) and *Dz*CYP94N8/*Dz*CYP94D143 (C26-oxidation), in *D. zingiberensis* were identified and transferred into the cholesterol-producing yeast (*Saccharomyces cerevisiae*) strain RH6829 to produce diosgenin successfully [10]. Based on those studies, a high-throughput function characterization platform was built in tobacco harboring a combination of *Dz*CYP90B71, *Dz*CYP90G6 and DzCYP94N8. Transgenic tobacco containing *Dz*CYP90B71 and *Dz*CYP90G6 can provide enough substrate for the C26-hydroxylase to biosynthesize diosgenin, and it was exploited as a chassis to test the potential C26-oxidation catalyzing activity of the *Dz*CYP72A candidates. Meanwhile, *Dz*CYP94N8 was used as the positive control. Using this strategy, a novel *DzCYP72A12-4* gene encoding C26-hydroxylase was identified from *D. zingiberensis*. Diosgenin was detected in the transgenic tobacco harboring *DzCYP90B71/DzCYP90G6/DzCYP72A12-4* genes and not in the transgenic plants harboring *DzCYP90B71/DzCYP90G6* or the transgenic plants harboring *DzCYP72A12-4*, suggesting that *DzCYP72A12-4* encodes a C26-hydroxylase involved in diosgenin biosynthesis.

Drought resistance can be understood as the ability of plants to sense the water-deficiency signal and initiate coping strategies, which are very complex traits that show up as diverse indicators for assessing improved drought resistance. The germination rate (GR) is considered as a common index of seed vigor. Seed germination usually needs sufficient water. Drought stress can result in decreased vitality and increased harmful substances, such as malondialdehyde (MDA). MDA is the final decomposition product of membrane lipid peroxidation, and the content of MDA can reflect the degree of stress injury suffered by plants. The transgenic tobacco plants (OE3) had a higher germination rate than the controls (NC1, NC2 and WT). Meanwhile, the MDA concentration in 16-day-old OE3 seedlings was lower than that in the controls. The emerging diosgenin could be the main cause of this phenomenon. A previous report showed that diosgenin has the ability to reduce oxidative stress damage [5]. The reconstitution of diosgenin biosynthesis in tobacco might lead to a promotion in drought adaptability. In addition, the expression level of *DzCYP72A12-4* was upregulated after treatments of PEG6000 (15%) in *D. zingiberensis*, further proving the relationship between *DzCYP72A12-4* and drought stress. However, the specific mechanism needs to be illustrated through further research.

## 4. Materials and Methods

### 4.1. Plant Materials and Growth Conditions

*D. zingiberensis* plants were collected from Ankang of Shaanxi province in China. Fresh rhizomes were transplanted into a greenhouse with strict environmental control (25 ± 2 °C and 16 h light/day). New plants sprouted and were propagated to produce seeds. Samples pending determination were freeze-dried to constant weight at −80 °C and ground into powder with a tissuelyser.

Tobacco (*Nicotiana benthamiana*) was used for the transgenic assays. Mature tobacco seeds were surface sterilized for 60 s with 75% (*v*/*v*) ethanol and rinsed with sterile water three times. Sterilized seeds were sown on 1/2 Murashige and Skoog (MS) medium (*Phyto Technology* Laboratories, Lenexa, KS, USA) containing 3% sucrose, and 0.75% agar, adjusted with 1 M KOH to pH 5.8. After two days of vernalizing at 4 °C, the seeds were germinated under the 16 h light/8 h dark photoperiod at 23 °C in a growth chamber. Materials of tissue culture were cultured in other chambers with the same condition.

### 4.2. Standards and Chemical Reagents

Chemical reagents of HPLC grade including methanol, ethanol, chloroform, acetonitrile and n-hexane were purchased from Thermo Fisher Scientific Inc. (Waltham, MA, USA). Standard chemicals such as dioscin (98%) and diosgenin (98%) were purchased from Shanghai Yuanye Bio-Technology Co., Ltd. (Shanghai, China).

### 4.3. Bioinformatics Analysis of CYP72A Genes

The basic local alignment search tool (BLAST) was used to search candidate CYP450 genes in the *D. zingiberensis* genome (NCBI project number: PRJNA716093) against functional characterized CYP450s (E-value 1 × 10^−10^, identity 50%). The amino acid sequence was analyzed using the online tool ExPASy (https://web.expasy.org/protparam/ accessed on 18 November 2022) [34]. Multiple sequence alignment was performed using the VectorNTI Advance software (version 11.5, Thermo Fisher Scientific, Frederick, MD, USA). The phylogenetic tree was constructed using MEGA11 by the neighbor-joining method [35]. Pairwise alignment was performed using the R Biostrings package [36]. The conserved motifs of *Dz*CYP72A12-4 proteins were analyzed using the following online tools:InterPro (http://www.ebi.ac.uk/interpro/search/sequence accessed on 15 November 2022) [37];SMART (http://smart.embl-heidelberg.de accessed on 15 November 2022) [38];NCBI Conserved Domain Database (https://www.ncbi.nlm.nih.gov/Structure/cdd accessed on 18 November 2022) [39].

### 4.4. Vector Construction and Agrobacterium Tumefaciens-Mediated Transformation

For stable expression in tobacco, the coding sequences of the selected genes were amplified by PCR from *D. zingiberensis* cDNA using the primers shown in Appendix A. For single gene expression, the purified PCR products were inserted into pUC57 plasmids on which the gene expression cassettes (CAMV 35S promoter-gene-Nos terminator or UBI1 promoter-gene-Nos terminator) were assembled using Gibson Assembly^®^ Master Mix—Assembly (E2611) (*NEW ENGLAND* Biolabs, Beijing, China). Two restriction endonuclease recognition sites were added flanking the cassette simultaneously, facilitating multi-cassette assembly. Single or multiple cassettes were digested by the corresponding restriction endonuclease and inserted into a modified pCAMBIA plasmid via the same restriction endonuclease digested cohesive ends. In summary, four expression vectors were constructed for the plant transformation listed in Appendix A.

All constructs were verified by sequencing and transformed into *A. tumefaciens* strain GV3101 by the electroporation method (Micro pulser, Bio-Rad, Los Angeles, CA, USA). The colonies were validated by PCR analysis. *A. tumefaciens*-mediated transformation for stable expression in tobacco was performed using a leaf-disc infection method. Aseptic seedlings of wild type tobacco cultured for about 30 days were used. The edge of the leaf was cut off, laid on solid MS medium and cultured at 25 °C for 2 days in the dark. After that, the leaves were immersed in the transformed A. tumefaciens prepared in advance for 10 min. The leaves with A. tumefaciens were co-cultivated in the dark at 25 °C continuously for two days. After the co-cultivation, the leaves were transferred to new solid MS medium and cultured at 25 °C in the 16 h light/8 h dark photoperiod. Calluses growing on the edge of the leaves were isolated and cultured on MS medium containing hygromycin (50 mg/L). The clump buds were induced after 20 days of culture. PCR was used to select the positive transgenic plants, which were cultured for further study. Transient expression in tobacco was conducted as previously described [9], with some modifications.

### 4.5. RNA Extraction and Gene Expression Analysis

Total plant RNA was isolated using the RNeasy plant mini kit (Qiagen, Valencia, CA, USA) following the manufacturer’s instruction with modifications. Potentially contaminated DNA was eliminated by treatment with DNase I (Takara, Peking, China). The RNA quality and concentration were determined using a NanoDrop 2000 spectrophotometer (Thermo Fisher Scientific, Wilmington, NC, USA). First-strand cDNA was synthesized using 1 μg of total RNA and the HiScript III 1st Strand cDNA Synthesis Kit (Vazyme, Nanjing, China) following the manufacturer’s instructions. Quantitative real-time polymerase chain reaction (qRT-PCR) was conducted following the manufacturer’s instructions of the SYBR Green PCR Master Mix (Applied Biosystems, Warrington, UK) using the CFX96 Real-Time PCR Detection System (Bio-Rad, Los Angeles, CA, USA). The primers used in this study are listed in Appendix A. The total RNA from young leaves, mature leaves, stems, flowers and rhizomes of *D. zingiberensis* were used to examine the tissue expression profiles of *DzCYP72A12-4*. To explore the effects of drought stress, young leaves were respectively harvested at 4 h, 8 h, 12 h, 24 h, 48 h, 72 h, 96 h and 120 h after treatment with 15% PEG 6000 for RNA extraction. The leaves of the normally watered plants were used as the control. The expression pattern of the *DzCYP72A12-4* gene under drought stress was analyzed. *DzGAPDH* was used as an internal control. The qRT-PCR was set up with the following thermal cycler conditions: initial denaturation preheating at 95 °C for 3 min, followed by 40 cycles of denaturation at 95 °C for 10 s and annealing/extension at 57 °C for 30 s. Biological and technical replicates were conducted three times to ensure the accuracy of the results. The 2^−ΔΔCT^ method was used for the analysis of the relative gene expression compared to the control [40].

### 4.6. Analysis of Diosgenin

Analysis of diosgenin in transgenic tobaccos and *D. zingiberensis* was performed following the method reported previously with modifications [10,11,25]. Briefly, 50 mg of the freeze-dried leaf powder was dissolved in 1 mL solvent (2:1 chloroform/methanol) and sonicated for 30 min followed by centrifugation at 12,000× *g* for 10 min. The above treatment of each sample was repeated three times. The supernatant was merged each time and filtered by microfiltration (0.22 m) pending determination. The content of diosgenin was determined using the ultra-performance liquid chromatography (Vanquish UPLC 689 system, Thermo Fisher Scientific, Bremen, Germany) tandem MS (Q Exactive Hybrid Quadrupole-Orbitrap Mass Spectrometer, Thermo Fisher Scientific, Bremen, Germany). A Hypersil GOLD C18 column 690 (2.1 mm × 100 mm, 3 m, Thermo Fisher Scientific, Bremen, Germany) was equipped on the UPLC system, and the temperature of the chromatography column was set to 40 °C. The flow rate of the mobile phases was 0.3 mL/min.

To determine the diosgenin content, the UPLC–MS/MS system described above was utilized. The mobile phases were eluent A (water with 5 mM ammonium acetate) and eluent B (methanol with 5 mM ammonium acetate). The gradient program was as follows: from 0 to 3 min, isocratic 65% B; from 3 to 25 min, linear gradient of 65–99% B; from 25 to 33 min, isocratic 99% B; from 33 to 33.5 min, 99–65%; and from 33.5 min to 35 min, isocratic 65% B. The Q Exactive MS was operated in positive target-SIM mode with the following parameters: probe heater temp, 350 °C; ion source, HESI-II; diosgenin precursor ion selection at 415.32 *m*/*z* in positive ion mode. The raw data were processed by XCalibur (Thermo Fisher Scientific, Bremen, Germany).

### 4.7. Drought Stress Treatment in Transgenic Tobacco

The seeds were sterilized and sown on solid ½MS media. After two days of vernalization at 4 °C, the seeds were further grown at 23 °C under long-day photoperiod conditions of 16 h light/8 h dark. After germination, the seedlings of the same size were transplanted into new MS medium and cultivated for 20 days. All seedlings were grown in the same environment, and tobacco plants of wild type (WT) were used as a control. Then, the 28-day-old transgenic and WT tobacco seedlings were transplanted into new trays filled with sterilized vermiculite and watered with freshly prepared Hoagland solution.

The tobacco seed germination test was performed according to the International Rules for Seed Testing. The seeds were laid in petri dishes (9 cm in diameter) underlaid with 2 layers of filter paper. Then, 2 mL 15%PEG-6000 was added to the test groups, while 2 mL distilled water was added to the control group and replenished every day. The growing conditions were controlled as described above. Each petri dish contained 100 seeds, and 3 replicates were set up. The number of germinated seeds was recorded every day, and the germination rate was calculated on the 16th day to determine the physiological indexes.

The germination rate (GR) was the number of germinated seeds on day 16. The concentration of malondialdehyde (MDA) in the fresh seedlings was measured after 16 days of growth. The MDA content was determined by the Plant MDA Assay Kit with TBA (Shanghai Yuanye Bio-Technology Co., Ltd., Shanghai, China).

### 4.8. Statistical Analyses

The qualitative information regarding diosgenin was collected from mass spectra. All experiments were performed for three independent biological repeats, and at least three technical repeats were set each time. The data were analyzed using the Student’s *t* test. Differences were considered statistically significant when *p* < 0.05 and extremely significant when *p* < 0.01, marked with * and **, respectively.

## 5. Conclusions

In the present study, a novel *DzCYP72A12-4* gene was screened through comprehensive methods. Diosgenin was detected in the transgenic tobacco plants (OE3) by UPLC–MS/MSv and not in the tobacco plants of the control groups lacking at least one of *DzCYP90B71*, *DzCYP90G6* or *DzCYP72A12-4*. In addition, the diosgenin-producing tobacco plants showed relatively high drought adaptability. All these findings could strengthen the understanding of the *DzCYP72A* gene subfamily in diosgenin biosynthesis and lay the foundation for the further exploration of drought resistance in *D. zingiberensis*.

Considering the large family of *Dz*CYP72A, further study should be conducted to confirm whether other *Dz*CYP72As are involved in diosgenin biosynthesis. In addition, the platform established in this study would be very useful to carry out further identification of *Dz*CYP72As as well as other genes of interest. Meanwhile, since steroid compounds have skeletons of a similar structure, it is worth exploring the structural similarities between key enzymes catalyzing similar substrates. In addition, the relationship between *Dz*CYP72A and drought stress deserves more attention. At present, although the diosgenin biosynthesis from cholesterol has been preliminarily clarified, more research is needed to reveal the full picture of the diosgenin biosynthesis pathway. Specifically, efforts should be made with respect to the following three aspects: (1) finding more key enzymes involved in diosgenin biosynthesis, (2) revealing the possible regulatory mechanism of diosgenin synthesis, and (3) analyzing the interaction effects of different bioactive compounds.

## Figures and Tables

**Figure 1 ijms-24-08430-f001:**
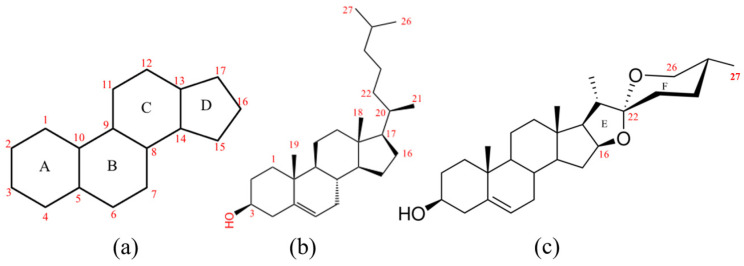
Chemical structure of the steroid ring skeleton, cholesterol and diosgenin. (**a**) Steroid ring skeleton. The Arabic numerals in red indicate the order of the skeleton carbon atoms. The capital letters represent different carbon rings. The basic steroid ring skeleton contains four carbon rings, A–D. (**b**) Cholesterol contains 27 carbon atoms, and the molecular formula is C_27_H_46_O. (**c**) Diosgenin. Two rings (E and F) are formed on the basis of the cholesterol. Its molecular formula is C_27_H_42_O_3_, and the relative molecular mass is about 414.32 calculated using Xcalibur (Thermo Fisher Scientific, Bremen, Germany).

**Figure 2 ijms-24-08430-f002:**
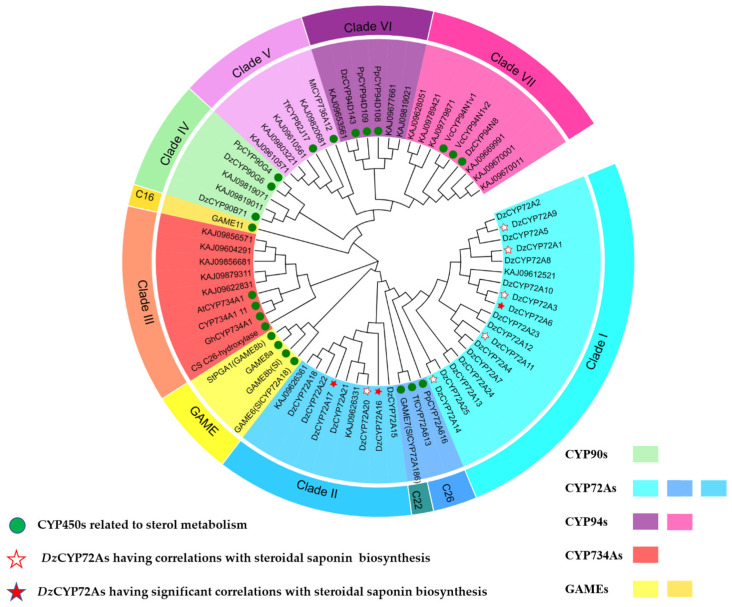
Phylogenetic tree of selected CYP450 proteins. Sequences with a green circle were identified CYP450s related to sterol metabolism; DzCYP72A sequences with a star were reported to have correlations with specialized metabolites of steroidal saponin biosynthesis, and a solid red star means a significant correlation [26].

**Figure 3 ijms-24-08430-f003:**
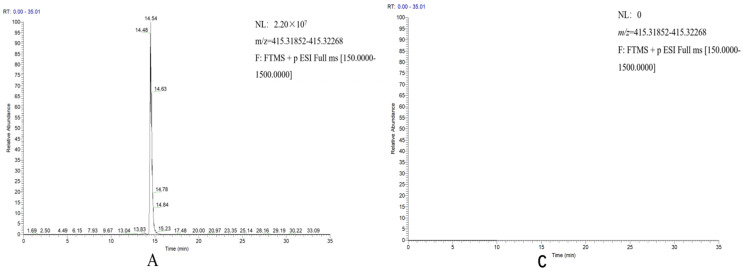
The ultra-performance liquid chromatography tandem MS analysis of the diosgenin in tobacco. (**A**) The samples extracted from *DzCYP90B71/DzCYP90G6/DzCYP72A12-4* transgenic tobacco plants. (**B**) The samples of standard diosgenin, which were used to illustrate the retention time and mass spectra. (**C**) The samples from *DzCYP72A12-4* transgenic tobaccos. (**D**) The samples from *DzCYP90B71/DzCYP90G6* transgenic tobaccos. (**E**) The samples from wild type tobaccos.

**Figure 4 ijms-24-08430-f004:**
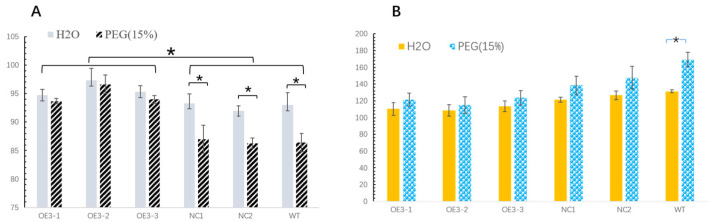
The germination rate and malondialdehyde (MDA) concentration of the transgenic tobacco plants under the stress of 15% PEG. (**A**) Germination rate. (**B**) MDA concentration. OE3-1/2/3, transgenic plants containing *DzCYP90B71/DzCYP90G6/DzCYP72A12-4*. NC1, transgenic plants containing *DzCYP72A12-4*. NC2, transgenic plants containing *DzCYP90B71/DzCYP90G6*. WT, wild type plants. The data were analyzed using the Student’s *t* test. Differences were considered statistically significant when *p* < 0.05, marked with *.

**Figure 5 ijms-24-08430-f005:**
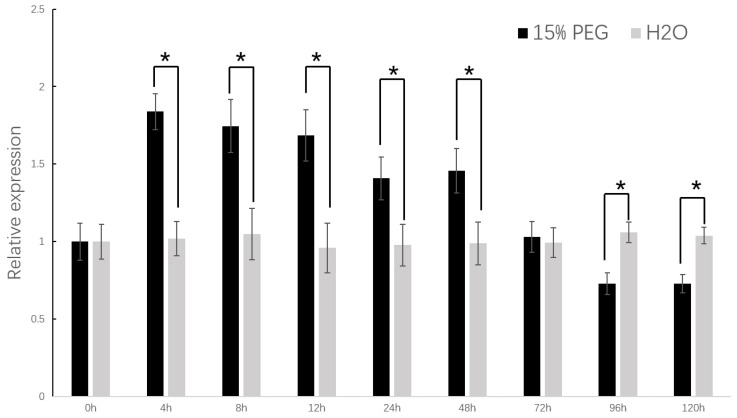
Analysis of *DzCYP72A12-4* gene expression. Relative expression of the *DzCYP72A12-4* gene under drought stress. H_2_O: control; 15% PEG: drought stress group. Relative expression levels, determined by qRT-PCR, relative to the expression of the *DzGDPH* gene. The data were analyzed using the Student’s *t* test. Differences were considered statistically significant when *p* < 0.05, marked with *.

## Data Availability

All study data are included in the main text and Appendix A.

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
