# Peer review of "The Functional Characterization of DzCYP72A12-4 Related to Diosgenin Biosynthesis and Drought Adaptability in Dioscorea zingiberensis"

_ijms, 2023, doi:10.3390/ijms24098430_

Round 1
Reviewer 1 Report
I have reviewed the paper titled “Functional Characterization of DzCYP72A12-4 Related to Diosgenin Biosynthesis and Drought Response in Dioscorea zingiberensis”. The paper contains a systematic record of scientific information. However, it needs editing before acceptance.
Comments
-The abstract should be more decorated.
- The introduction should be updated by adding related data from SCOPUS.
-The results and discussion should be improved.
- The resolution of figures 1 and 2 must be improved.
-The paper contains several typographical and grammatical errors which should be fixed.
Reviewer 2 Report
This is a nice piece of experimentation performed by Wang et al., which shows the Functional Characterization of DzCYP72A12-4 Related to Diosgenin Biosynthesis and Drought Response in Dioscorea zingiberensis"
Please check the types errors in the compelete manuscript before its final publication.
Authors should present their point of view on future perspectives.
Reviewer 3 Report
Please recheck the sentences to improve English.
Author Response
Thanks for the reviewer’s comments. We have revised the article carefully and corrected some grammatical and typographical errors one by one. In the meantime, the manuscript has been revised and decorated using the editing services listed at https://www.mdpi.com/authors/english.

Reviewer 4 Report
This is an interesting work, aiming to improve the production of diosgenin, a very important steroid for the industrial production of several steroidal drugs.
I consider that the work has impact and quality to be published.
Some points to be considered:
-I consider that the title of section “2.6. Analysis of Specialized Metabolites” is not much adequate, because only diosgenin was analyzed
-in addition, what about the validation of HPLC method to detect and quantify diosgenin?
-authors referred in section 2.8 “Data were analyzed using the Student’s t test. Differences were considered statistically significant when P < 0.05 and extremely significant when P < 0.01, marked with * and **, respectively.”. However, in figures, the asterisks to indicate significance are absent..
Round 2
Reviewer 1 Report
The paper should be accepted now.
Reviewer 4 Report
The document was improved.